



# A Distributed Temperature Profiling System for Vertically and Laterally Dense Acquisition of Soil and Snow Temperature

Baptiste Dafflon[1], Stijn Wielandt[1], John Lamb[1], Patrick McClure[1], Ian Shirley[1], Sebastian Uhlemann[1], Chen Wang[1], Sylvain Fiolleau[1], Carlotta Brunetti[1], F. Hunter Akins[1], John Fitzpatrick[2], Samuel Pullman[2], Robert Busey[3], Craig Ulrich[1], John Peterson[1], Susan S. Hubbard[1]

[1] Lawrence Berkeley National Laboratory, Berkeley, CA 94720, USA
[2] Independent, Oakland, CA 94501, USA
[3] University of Alaska Fairbanks, Fairbanks, AK 99775, USA

*Correspondence to*: Baptiste Dafflon (bdafflon@lbl.gov)

**Abstract.** Measuring soil and snow temperature with high vertical and lateral resolution is critical for advancing the predictive understanding of thermal and hydro-biogeochemical processes that govern the behavior of environmental systems. Vertically resolved soil temperature measurements enable the estimation of soil thermal regimes, freeze/thaw layer thickness, thermal parameters, and heat and/or water fluxes. Similarly, they can be used to capture the snow thickness and the snowpack thermal parameters and fluxes. However, these measurements are challenging to acquire using conventional approaches due to their total cost, their limited vertical resolution, and their large installation footprint. This study presents the development and validation of a novel Distributed Temperature Profiling (DTP) system that addresses these challenges. The system leverages digital temperature sensors to provide unprecedented, finely resolved depth-profiles of temperature measurements with flexibility in system geometry and vertical resolution. The integrated miniaturized logger enables automated data acquisition, management, and wireless transfer. A novel calibration approach adapted to the DTP system confirms the factory-assured sensor accuracy of +/– 0.1 ℃ and enables improving it to +/– 0.015℃. Numerical experiments indicate that, under normal environmental conditions, an additional error of 0.01% in amplitude and 70 seconds time delay in amplitude for a diurnal period can be expected, owing to the DTP housing. We demonstrate the DTP systems capability at two field sites, one focused on understanding how snow dynamics influence mountainous water resources, and the other focused on understanding how soil properties influence carbon cycling. Results indicate that the DTP system reliably captures the dynamics in snow thickness, and soil freezing and thawing depth, enabling advances in understanding the intensity and timing in surface processes and their impact on subsurface thermal-hydrological regimes. Overall, the DTP system fulfills the needs for data accuracy, minimal power consumption, and low total cost, enabling advances in the multiscale understanding of various cryospheric and hydro-biogeochemical processes.





## 1 Introduction

Temperature is a key property for understanding and quantifying a multitude of processes occurring in and across the deep subsurface, soil, snow, vegetation and atmosphere compartments of our Earth (e.g., Dingman, 2014; García et al., 2018). In addition to being a manifestation of thermal energy modulated by the heterogeneity of a given medium's thermal parameters, temperature influences a myriad of above- and belowground processes, including aboveground biological dynamics, energy-water exchanges, subsurface heat and water fluxes, soil and root biogeochemical processes, and cryospheric processes (e.g.,

Chang et al., 2021; Davidson and Janssens, 2006; Jorgenson et al., 2010; Natali et al., 2019). The predictive understanding of the above-mentioned processes across a large range of gradients in topography, air mass exposure, geology, soil type, and vegetation cover requires reliable measurement of the spatial and temporal distribution of snow and/or soil temperature (e.g., Lundquist et al., 2019; Strachan et al., 2016).

Time-series of soil temperature data have been used to improve understanding of a range of ecosystem properties and

processes. For example, temperature time-series have been used to explore the control that climate and subsurface properties have over permafrost dynamics (Brewer, 1958; Jorgenson et al., 2010), biogeochemical fluxes (Reichstein and Beer, 2008), plant function and root growth (Iversen et al., 2015), species and community distribution (Myers-Smith et al., 2011), and heat and water fluxes (Cable et al., 2014). Further, many studies have used temperature data to determine the vertical velocity of groundwater flow, surface-water–groundwater exchange, and groundwater recharge (Bredehoeft and Papaopulos,

1965; Briggs et al., 2014; Constantz, 2008; Hatch et al., 2006; Irvine et al., 2020; Racz et al., 2012). Time-series of temperature data have also been used in a parameter-estimation framework to quantify the soil thermal parameters and, in some cases, the fraction of soil constituents including organic matter content (Beardsmore et al., 2020; Nicolsky et al., 2009; Tabbagh et al., 2017; Tran et al., 2017; Zhu et al., 2019). Similarly, other studies have used vertically resolved temperature measurements in snow to infer snow thickness (e.g., Reusser and Zehe, 2011), snow thermal diffusivity (e.g., Oldroyd et al.,

2013), and improve the predictive understanding of snowpack dynamics in general (e.g., Reusser and Zehe, 2011).

Besides the acquisition of temperature time-series, soil temperature has been used to map temperature regimes by sequentially moving instruments to tens to thousands of locations across the landscape, down to a depth where thermal anomalies are larger than the effect of diurnal fluctuation. Early work was done by Cartwright (1968), who, using a thermistor at the end of an aluminum-tipped stake, found close agreement between locations of thermal anomalies and the

shallow aquifers that partly drove these anomalies. Furthermore, several successful studies have been performed in volcanic and hydrothermal areas to delineate thermal anomalies and in some cases calculate ground fluxes (e.g., Hurwitz et al., 2012; Lubenow et al., 2016; Saba et al., 2007). In a discontinuous permafrost environment, Léger et al. (2019) moved custom designed vertically resolved temperature probes sequentially across the landscape to identify near-surface permafrost distribution.

Design characteristics of devices sensing soil or snow temperature at multiple depths are largely driven by desires to jointly optimize cost, data information content and measurement consistency at each monitoring point, including numerous depths





and spatially distributed locations. Critical characteristics include measurement accuracy, autonomous data collection with high temporal frequency at a low power consumption, the ability to withstand rough environmental stresses and limit the disturbance of the sensed environment, small footprint, and total data cost (including material, deployment and management)

for duplicability. Several tools have been developed to provide vertically resolved temperature measurements at a limited number of spatially distributed locations, or at numerous locations but with poor vertical resolution. Examples of currently-available tools include : (1) a point-scale array of self-logging temperature sensors aligned inside a pipe (Constantz et al., 2002; Naranjo and Turcotte, 2015; Rau et al., 2010); (2) a point-scale array of thermocouple, thermistor, or digital sensors wired into a single electronic data-logging device (Cable et al., 2016; Constantz et al., 2002; Léger et al., 2019); and (3)

fiberoptic distributed temperature sensing that measures temperature at various locations and depths (Briggs et al., 2013; Vogt et al., 2010). While the cost of traditional temperature point sensors can be considered low (in the range of USD1 to USD 150), the total cost using the point-sensor methods—including the data logger, packaging, installation, localization, and management—can increase quickly and can limit extensive installation. Various efforts have concentrated on improving the packaging of sensors to ease data collection (Fanelli and Lautz, 2008; Gordon et al., 2013; Rau et al., 2010; Tonina et al.,

2014), still without fundamentally overcoming other limitations. Recent developments, including custom vertically resolved probes linked to commercial (Aguilar et al., 2018; Andújar Márquez et al., 2016; Naranjo and Turcotte, 2015) or in-house loggers (Beardsmore et al., 2020; Léger et al., 2019), as well as some commercially available systems, are still limited in their vertical resolution, flexibility, and cost effectiveness for wide deployment. While fiber optic-based methods have been widely applied for temperature measurement in deep wells, infrastructures, and streambeds (Briggs et al., 2012), their

deployment for numerous shallow and vertically resolved depth profiling of temperature is still challenging. This is due to the high initial investment required and the risk of losing a large amount of data in the case of instrument, cable, or power failure (Lundquist and Lott, 2008). Finally, it can be noted that the absence of systems to efficiently map soil thermal regimes at hundreds of locations has been recognized by several studies that have either relied on conventional thermocouple probes (≤25 cm) (e.g., Leon et al., 2014; Lubenow et al., 2016; Price et al., 2017) or developed their own acquisition devices

that are costly to duplicate (Hurwitz et al., 2012; Léger et al., 2019).

Spatially and temporally dense monitoring of temperature –acquired at hundreds of locations distributed across several square kilometers– have primarily focused on air (Alcoforado and Andrade, 2006; Holden et al., 2016; Hubbart et al., 2005; Whiteman et al., 2000) and ground surface temperature (Davesne et al., 2017; Gisnås et al., 2014; Gubler et al., 2011; Lewkowicz et al., 2012).  For example, dense air temperature measurements from 1600 self-recording temperature sensors,

across a domain extending from the Boise Basin, Idaho, to southern British Columbia, have been used to evaluate downscaling of air temperature from long-term weather stations – using covariates that have established physical links to surface air temperature (Holden et al., 2016). In another study, 171 sensors recording the distribution of ground surface temperatures, across a climatic gradient from continuous to sporadic permafrost in Norway. documented the pronounced control of snow depth over the local-scale variability in mean annual ground surface temperature (Gisnås et al., 2014).



Acquisition of depth-resolved soil temperature at hundreds of locations has been rare and generally the result of long-term efforts or coordinated collaborative efforts (Cable et al., 2016; Nelson et al., 1998; Shiklomanov et al., 2008)

Mapping or monitoring depth-resolved profiles of soil or snow temperature, and the scientific insights anticipated from data having much higher spatiotemporal resolution than currently possible, requires advances in flexible, affordable, and community-available temperature profiling systems, with custom hardware, software, and packaging, enabling optimized

power consumption, accuracy, resolution, data transfer, and data cost management. In fact, while the "V's" (velocity, volume, variety, value, and veracity) scores (Demchenko et al., 2013) of temperature measurements in a "Big Data" era are presumably very high in comparison to other measurements, there is room for significant improvements. This potential is mainly a result of recent advances in semiconductor technology, allowing miniaturized digital temperature sensors with an unprecedented cost, accuracy, resolution, stability, and power consumption. Increasing the temperature "V's" for mapping

and monitoring soil or snow temperature in the earth sciences promises to improve our ability to capture ecosystem dynamics across a large range of gradients in landscape properties. "V's improvement" would in turn improve data- or model-based prediction of heat and water fluxes at multiple scales, reduce uncertainty in prediction of biogeochemical processes influenced by thermal and hydrological regimes, and move the community toward near-real-time predictions of hydro-biogeochemical processes using data streamed from the field. While recent technological advances in low-cost and

low-power digital sensors facilitate the development of inexpensive and customizable platforms, including sensors and loggers, microcontrollers, and communication modules, efforts are still needed to integrate low-cost sensors and loggers for increasing spatial coverage and facilitating new insights into environmental process dynamics.

The objective of this study is to design and develop a distributed temperature profiling (DTP) system for characterizing and/or monitoring vertically resolved profiles of snow and soil temperature at an unparalleled number of locations, to

improve our predictive understanding of hydro-biogeochemical ecosystem processes. In particular, this development is aimed at advancing snow or soil temperature measurements at multiple locations for various purposes, including (1) quantifying snow thickness and snowpack dynamics, (2) inferring soil thermal metrics (e.g., thaw layer thickness), (3) estimating soil thermal parameters and/or heat/water fluxes using data and physically based models, (4) developing proxies to facilitate the transfer of knowledge from intensive but sparsely distributed sites to sites where only a subset of variables

are measured, and (5) integrating ground-based data with remote sensing products for improved mapping of hydro-biogeochemical properties. To potentially fulfill the above goals, we hypothesize that measuring soil and snow temperature with unprecedented vertical and lateral resolution and relatively high accuracy (<0.05°C) can become feasible with the development of a novel DTP system. Although an earlier prototype of a DTP system (Léger et al., 2019) offered a new paradigm in sequentially acquiring vertically resolved soil temperature measurements across the landscape, its limited

accuracy of 0.15°C, the time required to assemble the system, and the high power consumption and footprint of the connected Raspberry PI based logger limited its wide applicability.

In this study, we designed and field-tested a DTP system that enables (1) customized deployment of probes with flexibility in assembling systems of different length, housing, vertical resolution, and accuracy, depending on the subsurface phenomena



being sensed; (2) durability, specifically the ability to withstand rough environmental stresses, and (3) dense acquisition of
measurements by minimizing total cost (including the costs of material, construction, deployment, and data management)
and device footprint. An additional important step in this study for limiting device cost and footprint is the design of a
miniaturized, low-power logger with wireless connectivity for downloading data and setting up acquisition parameters,
allowing for possible future integration within a LoRa wireless sensor network (Wielandt and Dafflon, 2020). In the
following, we first describe the design and components of the newly developed DTP system, providing sufficient detail for
others to build a DTP system. Then we present a new, lab-based calibration approach to assess and, if desired, improve the
DTP sensor accuracy. In addition, we assess the specifics of the developed system using numerical modeling and we
demonstrate its applications in two field cases—to measure snow and soil temperature, and to infer snow thickness and soil-
thawed and frozen-layer thickness. Finally, we discuss the system's advantages and limitations.

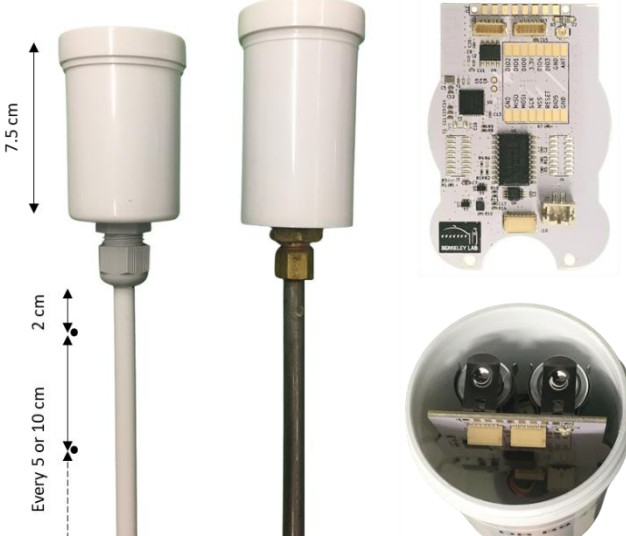


**Figure 1: General overview of the DTP system. The DTP system can be assembled in various lengths with temperature sensors
every 5 or 10 cm along the probe, and packaged in a plastic or steel tube, depending on deployment goals and environmental
conditions. The data logger controls the temperature sensors on the probe, sequentially reading and storing the temperature data.
An Android app is used to communicate with the logger and download data wirelessly.**

**2 Method**

**2.1 DTP system hardware and connectivity**

We designed a system composed of digital temperature sensors mounted on an array of cascaded Printed Circuit Board
(PCBs) connected to a custom-designed low-power logger. The sequentially addressable digital temperature sensors
(TMP117AIDRV) are low-cost, low-power, and high-accuracy, with a resolution of 0.0078125 ℃ and a factory-assured,
National Institute of Standards and Technology (NIST) traceable accuracy of +/- 0.1 ℃ across a temperature range of -20℃





to 50°C (http://www.ti.com/lit/ds/symlink/tmp117.pdf). All sensors on the probe are connected to the data logger's Two Wire Interface (TWI), and each sensor is accompanied by a discrete D flip flop that propagates an address bit along the probe. This approach enables a sequential readout of an arbitrary number of sensors with just six signals (3.3 V supply, Ground, TWI clock, TWI data, address, address clock). The board-to-board connections between probe sections rely on

custom-designed press-fit PCB connectors to ensure lasting structural stability and electrical contact under mechanical and thermal stress in the field. Once assembled, each PCB section is 20 cm long and contains 2 or 4 temperature sensors to enable 10 or 5 cm spacing, respectively. The upper PCB section is connected to a press-fit wire-to-board adaptor to link the entire probe assembly to the logger. The electrical design of the boards minimizes capacitive loading and crosstalk of the communication signals. In combination with a TCA9803 TWI bus buffer, this allows for sensor arrays over 2 m long,

without affecting signal integrity. The entire temperature probe is powered down in between measurements, resulting in a 0.0 µA idle current and a reduced impact of electrical failures along the probe. A measurement of 16 sensors along the probe takes 100 ms and requires up to 220 µA per TMP117AIDRV.

The logger is a custom-designed embedded system built around a low-power wireless system-on-chip (NRF52832 ARM Cortex M4) that enables Bluetooth Low Energy (BLE) connectivity. On-board provisions include a TCA9803 TWI buffer, a

load switch, a TMP117AIDRV temperature sensor, 32 Mb of low-power NOR FLASH memory for storing measurement data, a temperature-compensated real-time clock (RTC) for accurate time keeping and generating watchdog and measurement interrupts, multiple connectors for existing and future sensor expansion, and an RFM95W LoRa modem for future integration in LoRa wireless sensor networks (Wielandt and Dafflon, 2020). The system operates in the 1.8 V to 3.6 V range, allowing dual AA battery operation without requiring further power supply circuitry. The microcontroller and its

peripherals are mostly asleep, drawing a system idle current of 7.085 µA. Taking regular BLE advertising and a 15-minute sensor-measurement interval for 16 sensors into account, the total system's current consumption averages at 22 µA. Using Energizer L91 batteries (https://data.energizer.com/pdfs/l91.pdf) with a 3500 mAh capacity, a total battery lifetime of 18 years can be reached in theory. With the above settings, the on-board flash memory would be filled in 3 years.

Logger parameters (measurement interval, time, etc.) and on-board stored data are managed using BLE connectivity and a

custom companion app for Android devices. The app provides a list of nearby probe identifiers ranked by their Bluetooth signal strength, which usually correlates to the distance from the Android device. The app allows a user to erase logger memory, reset the system, synchronize the on-board clock, set a logging interval, transfer data, and assign GPS coordinates through the phone's GPS. Current data transfer speed is ~60 kb/s, which means that two weeks of data are downloaded every second (assuming a 15 min sampling interval and 16 connected sensors). The transferred data are converted into a .csv

format.

## 2.2 DTP system assembly and deployment

The probe is built by cascading sensor boards to the desired length and inserting the sensor assembly into a tube, which is then further filled with a sealing urethane mixture and connected to a logger and its enclosure. Different types of tubes and





connections to the logger can be used, based on the application. The default tube is a ⅜" outer diameter (OD), ¼" inner
diameter (ID) white cellulose acetate butyrate (CAB) plastic tube that is easily machinable, flexible, UV-resistant, high-
albedo, and structurally stable in cold and warm temperatures. Alternatives include a ⅜" OD, ¼" ID 304 stainless steel tube,
or a ½" OD, ¼" ID CAB tube.

The tube is cut to fit the desired length of the connected PCB sections, with up to 5 cm excess to accommodate for the tail
end of the final PCB. A cable gland is tightened and –if necessary– glued on the top of each tube. Then, the tube is filled
from the bottom with a urethane blend using a syringe to reduce the chance of air bubbles. The urethane blend (20-2360
from Epoxies Inc., https://www.epoxies.com/_resources/common/bulletins/20-2360R.pdf) is a thermosetting mixture
designed for electrical potting applications over a temperature range of -40 to 125ºC, and has a measured thermal
conductivity of 0.191 W/mK. Its coefficient of thermal expansion (2.28e-4) and high tensile strength (400 PSI) limit the risk
for the probe to warp or snap under a large range of thermal or physical conditions. A ⅜" OD metal spike is added at the
bottom of the tube to act as a stopper while the urethane mixture sets, ease ground entry during deployment, and enable some
electrical grounding with the use of a grounding wire attached at the bottom of the sensor assembly. A 4 oz. polypropylene
(PP) jar can be mounted to the top of the probe and serves as a UV-resistant, dust and splashproof enclosure for the data
logger (Figure 1). Depending on the application, sealant can be applied on the jar seams to achieve long-term waterproofing
and submergibility.

The cost of materials for the default DTP system, including the logger, can be as low as $95 USD for a 1.2 m long probe
with 16 temperature sensors, assuming a batch size of ~300 probes. The cost is distributed between the logger components
and manufacturing ($19 USD), the batteries ($2 USD), the CAB tube ($4 USD), the logger enclosure ($1 USD), the urethane
mixture ($2 USD), the cable glands ($2 USD), and the sensor boards ($65 USD) which include the cost of $2.5 USD per
TMP117 sensor. A 304 stainless steel probe implies an additional $20 USD, distributed between the stainless-steel tube ($10
USD) and the brass tube fitting ($10 USD). These price estimates do not take into account the cost of the mechanical
assembly of the various sensor boards and logger into their final housing. In addition, the above-mentioned cost of the logger
and sensor boards is only obtained under optimal factory yields and strongly depends on choices and fluctuations in
component and PCB manufacturing prices. In sub-optimal conditions, additional costs can easily add up to ~$120 per probe.

The field deployment of the DTP system can be performed in various ways, depending on the probe housing and application.
For plastic probes, a custom-length drill bit with the same diameter of the probe is used to drill a guide hole in which the
probe is then inserted. The probe can be inserted completely into the ground, or part of it can be left above ground (Figure 1).
Stainless-steel probes have the ability to be directly hammered into the soil if the environmental conditions enable it.
Aboveground installation of the DTP system for snow temperature measurement is done by attaching the probe to a PVC or
wood stake using low-temperature-resistant zip-ties or tape attached at mid distance between consecutive temperature
sensors.



## 2.3 DTP sensor accuracy assessment and calibration

A procedure was developed to evaluate the accuracy of a sensor marketed with a factory-assured NIST traceable accuracy of +/- 0.1℃, and possibly to improve its accuracy with an additional calibration procedure. The most common method for calibrating a temperature sensor consists of a single point calibration where a sensor is submerged in an ice bath, made by

saturating 2-3 mm particles of shaved or crushed ice in distilled water, and allowing the mixture to equilibrate (Mangum, 1995). If carefully prepared, the latent heat of fusion, which is needed for the phase change, stabilizes the bath within a few ten-thousandths (~0.01) of 0℃ (Thomas, 1938). Cable et al. (2016) used this calibration approach to increase the accuracy of thermistors from 0.1℃ to approximately 0.02℃ for subsurface temperature measurements. While temperature-controlled water baths at temperatures above 0℃ (Aguilar et al., 2018; Naranjo and Turcotte, 2015) can be used for calibration using a

reference thermometer, reaching an accuracy of 0.01℃ is challenging.

While the standard ice-bath approach is adapted for calibration of individual sensors or a string of sensors, initial tests performed in this study did not provide satisfactory results when scaling up this approach to submerge an entire 1.2 m long DTP system. Initial tests were conducted by filling a 1.5 m long 25 cm diameter pipe with a mixture of cool distilled water and cold crushed ice. The DTP system was centered in the pipe with a Fluke reference thermometer (Fluke1524) collocated

next to one of the DTP sensors for additional comparison. Results have shown that building a fine mixed water-ice bath at that scale was time-consuming, not always successful because of the difficulty of having a well homogenized mixture in such a large volume, and thus not adequate for calibrating hundreds of DTP systems.

In this study, a novel 0℃ point calibration approach was developed to calibrate tens of probes in one single run, while achieving accuracy similar to the ice-bath method. Our approach includes cycling through water-ice phase changes in a 1.5"

diameter tube filled with cold distilled water and with the DTP probe suspended at its center. Multiple probes and tubes are placed into an incubator (Thermo Scientific Precision Incubator) at -5℃ over a period of 12-24 hours to ensure frozen conditions, and then slowly warmed in an incubator at 3℃ until melting is complete. At the melting point, the measured temperatures flatten off due to the latent heat of fusion (heat-induced zero-curtain), and a value can be obtained for each individual sensor's offset from 0℃.

## 2.4 Assessment of controls on heat transfers

Numerical experiments were performed to evaluate the impact of various DTP characteristics and environmental factors on DTP measurement accuracy, beyond the sole sensor accuracy. In particular, we investigated how the temperature measurements are potentially impacted by the probe tubing material and diameter, different ground and probe surface heating, the air gap between the probe and soil, sensor positioning error, and variable soil thermal diffusivity.

A finite volume numerical model on an axis-symmetric cylindrical grid was developed to simulate heat transfer through conduction, in and between the probe and the soil. The thermal conductivity and heat capacity were explicitly represented in the model. Temperature in each cell was updated in time by summing the contributing heat flow across each cell boundary,





and stability was controlled by heuristically reducing the timestep to between $10^{-4}$ and $10^{-1}$ seconds. The model spanned across 50 cm and 50 cells vertically and 5 cm and 100 cells radially, and was parameterized with the thermal conductivity and heat capacity of the probe and soil. The initial conditions and the moving Dirichlet boundary conditions at the top and bottom were calculated using the analytical solution for diurnal heat transfer in the half plane (Turcotte and Schubert, 2002). Boundary conditions at the outside edges were similarly obtained with the analytical solution using the ghost point method. The internal boundary condition at zero radius was treated as a zero-flux Neumann boundary for symmetry (Petter Langtangen and Linge, 2017, p251). Validation of the numerical model was carried out by applying a naive T=0 Dirichlet boundary condition at the outside and bottom of a homogeneous domain, and a sinusoidal forcing function at the upper surface. The simulated temperatures closely matched the analytical solution. For the numerical experiments, the simulated temperatures inside the probe using the finite volume model were compared to the analytical solution. The differences were evaluated through the percentage mismatch and time delay in amplitude.

## 2.5 Autonomous estimation of soil and snow properties

Vertically resolved measurements of soil and temperature can be used to infer various properties, including soil thermal parameters, snow thickness, zero curtain duration, first bare-ground date, frozen and thaw layer thickness, and many empirical indices. In this study, we evaluate the value of the DTP system to autonomously estimate snow thickness, soil frozen, and thaw layer thickness, as well as the possible probe upward displacement relative to soil surface that can occur in frost-affected soil using acquired temperature measurements.

### 2.5.1 Snow thickness

The snow thickness can be estimated from a vertically resolved temperature probe placed above the ground surface by identifying where the maximum reduction of the diurnal temperature variation occurs along the vertically resolved profile (Oldroyd et al., 2013; Reusser and Zehe, 2011). This maximum reduction occurring at the air-snow interface is caused by the insulating effect of the snow. In this study, we use a numerical approach relatively similar to the one presented in Reusser and Zehe (2011). Reusser and Zehe (2011) demonstrated their approach by placing nine Hobo pendant temperature data loggers on a square metal rod with a spacing of 15 cm covering a range from 0 to 120 cm above ground. Deploying this instrumentation at five locations, they found that the resulting time series of snow height was in good agreement with their reference measurements done using ultrasonic sensors. The mean absolute error between both types of measurements was 6 cm, which corresponds to the expected minimum error for their setup where the temperature sensor spacing was 15 cm.

Our algorithm to retrieve snow thickness from the DTP system consists in (1) calculating the gradient in temperature between each pair of consecutive sensors along the probe at each sampling time (15 minutes per default), (2) disregarding pairs where both members indicate one or more temperature measurements > 2°C during a 24-hour window centered around the sampling time or where the temperature range is larger in the lowest sensor of the pair, (3) selecting the pair with the maximum range in gradient over the 24-hour window and assigning the snow depth estimate to the lowest sensor in the pair,



and (4) selecting only the solution where the obtained snow thickness corresponded to the mode value in the preceding or following 6 hours. The second and fourth steps are intended to avoid the possible occurrence of isolated suspicious estimates when the temperature diurnal variation is very small. The developed approach is relatively similar to the one presented by Reusser and Zehe (2011), with the major difference being that they relied on the maximum change in standard deviation over depth instead of the maximum range in gradient.

### 2.5.2 Frozen and thawed layer thickness and probe heave

Frozen and thawed layer thickness can be inferred from vertically resolved temperature measurements by extracting the 0℃ isotherm in the temperature time-series during the freezing and thawing period, respectively. The accuracy of the estimated frozen or thawed layer thickness depends on the vertical resolution of the DTP probe, the true freezing point of the material, the accuracy of the temperature measurement, and the positioning of the DTP probe relative to the soil surface. The possible movement of the probe relative to the ground surface over time, which can result from soil mechanical processes or animal disturbance, is obviously the source of uncertainty that is the most difficult to assess. For example, a common concern in the Arctic is that the sensor, stake, or probe can rise upward relative to the soil surface elevation, due to frost jacking or soil frost heave or thaw settlement processes (Iwahana et al., 2021; Johnson and Hansen, 1974; Matsuoka, 1994). This potential upward displacement of the object or material in freeze/thaw cycles depends on various environmental factors, is difficult to predict, and cannot be fully dismissed unless the instrumentation is anchored in bedrock or in cold permafrost. Though not investigated here, modifying the probe frictional surface could possibly minimize probe heave or frost jacking.

In this study, we evaluate the detection of possible probe displacement relative to the soil surface. To this end, we consider the time delay between diurnal fluctuation in temperature observed by the top sensor located above the ground surface and the other sensors initially located in the ground. The algorithm involves (1) filtering the dataset with a 1-hour moving window centered on each measurement, (2) selecting days when the aboveground sensor temperature shows a daily diurnal range in temperature larger than 4℃ and a maximum temperature higher than 0.1℃, (3) selecting sensors which when compared to the above-ground sensor show less than a 2℃ difference in their diurnal range in temperature and a time delay in minimum daily temperature of 15 minutes or less, and (4) defining an upward movement when the above difference and shift is observed for two consecutive days. Note that only considering the days when the top sensor above the ground surface shows a maximum temperature above 0.1℃ is intended to dismiss days when the top sensor is under the snow surface, which complicates the detection of upward movements. Overall, this detection method provides an initial approach for assessing probe displacement without visual inspection, as well as flagging or correcting temperature measurements and inferred metrics.





## 3 Results

### 3.1 Sensor accuracy assessment and additional calibration

The developed sensor accuracy assessment approach was validated by repeating the approach several times with one DTP system, and then applying the approach on 846 sensors from 70 probes (Figure 2). The zero-curtain induced by the water-phase change is consistently observed around 0°C, with offsets that are always smaller than the +/- 0.1°C factory-assured accuracy. Repeating the calibration cycle three times with the same probes shows that the offset of each sensor across the calibration cycles varies over a range of 0.015°C. The offsets of 846 sensors indicate a relatively gaussian distribution of offsets with a mean of +0.02433°C, a standard deviation of 0.02095°C, and a 95% percentile interval between -0.022°C and 0.062°C.

The results of the sensor accuracy assessment indicate that the sensor accuracy can be improved using the developed approach from a +/- 0.1°C factory-assured accuracy to about +/- 0.015°C. Note that the maximum offset measured on 846 sensors was +0.077°C, which indicates the already high accuracy of the factory calibration for the tested sensors. The only observed caveat for the calibration approach is that the top sensor along the probe does not always show the clear zero curtain needed for precise calibration, because it is not consistently covered with ice. This issue results from the need to leave sufficient air space, with some safety margin, between the water surface and the bottom of the logger, in order to account for ice extension, which is needed to avoid ice pushing on the logger directly and breaking the probe.

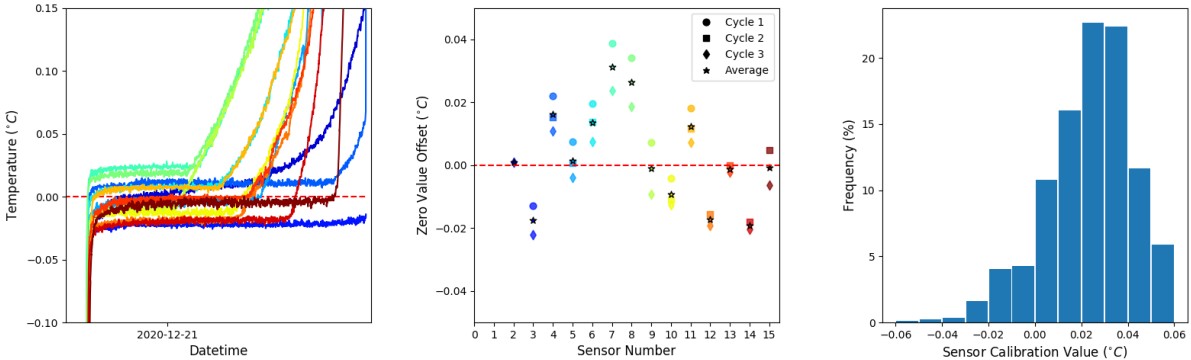

**Figure 2: Sensor accuracy assessment. (a) 0oC curtain occurring during ice-to-water phase transition and observed by each sensor along the DTP probe, with offset related to sensor factory accuracy; (b) offset values observed by running the assessment approach three times with the same probe, indicating that the additional calibration improves the accuracy of the sensor to +/- 0.015oC (i.e., based on variations in three cycles), (c) distribution of the sensor factory offsets obtained from 846 sensors from 70 probes, with a mean of 0.02433°C and a standard deviation of 0.02095°C.**





## 3.2 Numerical experiments

The effect of different probe characteristics and environmental factors on the measured temperature accuracy is quantified using numerical experiments. Accuracy is evaluated through the relative difference and time delay in diurnal amplitude between the hypothetically measured and true temperatures (Table 1 and Figure 3). The maximum percentage error between the measured and true temperature at all times is also considered, through their absolute difference divided by the maximum amplitude at the same depth. Note that sensor accuracy is not considered in these numerical experiments and thus needs to be

added to calculate the maximum total error.

The developed DTP system in its standard plastic housing (3/8'' OD plastic tubing filled with urethane blend) senses temperature with a maximum difference of -0.11% between the measured and true amplitude, and a time delay ranging between 45 and 95 seconds for a soil diffusivity between 0.15 and 1 mm$^2$/s, and assuming an absence of air gap between the probe and the soil. The maximum difference in amplitude is as small as 0.01% with a soil diffusivity of 0.5 mm$^2$/s.  The

temperature maximum error at any time and for the full range of soil diffusivity is ~0.7 %.

Increasing the diameter of the plastic probe to ½ '' OD, or considering a hypothetical 1 mm air gap, produces a time delay of up to 160 seconds in a soil with a diffusivity of 0.5 mm$^2$/s. The amplitude and measurement error can be as high as 0.03% and ~1.2%, respectively. While the error is almost double the standard case, it is acceptable for many applications. Still, results show that further increasing the probe diameter or the air gap increases inaccuracies significantly. For example, the

presence of an extreme hypothetical 5 mm air gap produces a time delay of up to 505 s (Figure 3).

Different surface heating between the soil and above-ground probe surface, though difficult to assess because of the complexity of the surface energy exchanges, is primarily influencing the surficial soil (top 5 cm) temperature measurements. A temperature difference between the probe and soil surface equal to half the diurnal variation can create an amplitude difference of ~6% at 1 cm depth, decreasing to less than 0.03% at depth deeper than 5 cm. Such different heating or cooling

responses between the soil and probe surfaces can result from different surface emissivity, insolation, near-surface wind, and water-phase changes in the soil. Deployment of probes at locations where environmental factors may strengthen this source of error could benefit from burying the probe and the logger separately, and using thin diameter plastic probes.

The use of stainless-steel housing may cause a slightly reduced accuracy in soil temperature measurement compared to the standard plastic 3/8'' OD tubing. Stainless-steel, which facilitates vertical heat transfer along the probe, results in a negative

time delay of 235 s maximum and a max error of 0.1% between the measured and true amplitude. The stainless-steel standard probe setting increases the potential for an overestimation of the *in situ* amplitude in the top 15 cm and then an underestimation similar to the plastic standard case. Though stainless-steel tubing limits the accuracy in the top part of the soil, overall it can provide a tighter contact with soil because a guide hole is not always required for inserting the stainless-steel probe. Stainless-steel tubing with no air gap has the potential to provide relatively comparable performance to a plastic

probe with an air gap larger than 2 mm. Finally, it can be noted that the use of aluminum instead of stainless-steel is inadequate, because it strongly decreases the measurement accuracy (Table 1).



The effect of most characteristics and factors mentioned above is minor compared to the error resulting from possible inaccuracies in positioning the sensor at a specific depth, which can occur with all measurement methods. Here, a hypothetical 1 cm downward shift of the probe can lead to an amplitude and measurement error of ~8% and ~12%,

respectively, and a time delay of 1244 s. These errors are two times larger than the effect of an air gap of 5 mm between the soil and the probe.

For the case where the DTP system is installed temporarily for capturing a single time or snapshot of the soil temperature for mapping purpose, the amount of time needed to approach temperature equilibrium between soil and sensors depends on environmental factors and desired measurement accuracy (Figure 4 and Table 2). The DTP system in its standard plastic

housing (base case; 3/8'' OD plastic tubing filled with urethane blend) and the stainless-steel probe require 824 and 1040 s, respectively, to reach 1% of the initial difference of temperature between the probe and the soil. Results indicate that a 1 mm air gap produces a significant delay in the early time of the equilibration process, although it reaches 1% of the initial difference after a comparable amount of time, i.e., 1070 s. In the presence of low soil diffusivity, the equilibration time increases to 1748 s, implying that leaving a probe in place for about 30 min is appropriate for many applications. Finally,

results indicate again the importance of ensuring a good coupling between the probe and the soil, as seen by the effect of a hypothetical 5 mm gap between the probe and soil, which more than doubles the equilibration time needed for reaching similar accuracy.

**Table 1: Evaluation of the impact of various factors on the measured temperature accuracy. The parameters for the base-case (Bc) scenario are changed one at a time to simulate various cases (Figure 3). The soil conductivity k was taken as a linear function of soil diffusivity a. The probe is filled with a urethane blend mixture (a=0.11 mm2/s, k= 0.204 Wm−1K−1). Along with the base case of a 10 mm plastic probe, different error-causing variations were simulated, including probe diameter variation by ±50%, a range of soil a, gaps of air between the probe and the soil, differential heating of the probe surface by ±50% of diurnal variation, a shift in the probe of 1 cm downward, and different probe casing, including stainless steel and aluminum.**

| Case | Simulation settings | | | | | | Simulation results | | | | | |
|---|---|---|---|---|---|---|---|---|---|---|---|---|
| | Material ($a$ (mm²/s), $k$ (Wm⁻¹K⁻¹)) | Probe diameter (mm) (OD, ID) | Air gap (mm) | Surface T diff. (°C) | Soil ($a$ (mm²/s), $k$ (W m⁻¹ K⁻¹)) | Shift in depth (mm) | Max Error in A (%) | Max Error in A (%) at z > 5cm | Max time delay (s) | Max time delay (s) at z > 5cm | Max relative error in T (%) | Max relative error in T (%) at z > 5cm |
| Base Case | (0.11, 0.204) | (10, 6) | 0 | 0 | (0.5, 1.4378) | 0 | -0.01 | -0.01 | 70.03 | 65.01 | 0.52 | 0.48 |
| Thin Probe | Bc | (5,3) | Bc | Bc | Bc | Bc | 0.00 | 0.00 | 34.96 | 20.04 | 0.24 | 0.13 |
| Thick Probe | Bc | (15, 9) | Bc | Bc | Bc | Bc | -0.03 | -0.03 | 160.19 | 160.19 | 1.16 | 1.16 |
| High Soil $a$ | Bc | Bc | Bc | Bc | (1, 2.5818) | Bc | 0.00 | 0.00 | 60.10 | 60.10 | 0.44 | 0.42 |
| Low Soil $a$ | Bc | Bc | Bc | Bc | (0.15, 0.6371) | Bc | -0.11 | -0.11 | 94.76 | 45.29 | 0.69 | 0.33 |
| 1mm Airgap | Bc | Bc | 1 | Bc | Bc | Bc | -0.01 | -0.01 | 145.00 | 145.00 | 1.06 | 1.03 |
| 5mm Airgap | Bc | Bc | 5 | Bc | Bc | Bc | -0.12 | -0.07 | 505.00 | 505.00 | 3.64 | 3.64 |
| Heated Top | Bc | Bc | Bc | 0.5 | Bc | Bc | 5.93 | 0.02 | 70.03 | 65.01 | 5.93 | 0.47 |
| Cooled Top | Bc | Bc | Bc | -0.5 | Bc | Bc | -5.93 | -0.03 | 105.14 | 70.03 | 5.98 | 0.49 |
| Shift 10 mm | Bc | Bc | Bc | Bc | Bc | 10 | -8.18 | -8.18 | 1244.99 | 1240.19 | 11.91 | 11.88 |
| Stain. Steel | (4.2, 16.2) | Bc | Bc | Bc | Bc | Bc | 0.10 | 0.05 | -235.01 | -235.01 | 1.69 | 1.69 |
| Aluminum | (69, 167) | Bc | Bc | Bc | Bc | Bc | -2.16 | -2.16 | -2800.00 | -2800.00 | 20.23 | 20.23 |


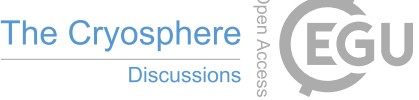

**Figure 3:** Influence of various probe characteristics and environmental factors on the accuracy of the DTP soil temperature measurements at various depth. (a) relative error in amplitude, (b) time delay in amplitude, and (c) measurement maximum percentage error relative to true amplitude at each depth.

**Table 2:** Time (in seconds) needed for the DTP sensor to approach soil temperature of 0.1, 0.05 and 0.01 times their initial differences, depending on various probe and environmental parameters. The parameters for the base-case (Bc) scenario are changed one at a time to simulate various cases (Figure 4). Table 1 provides the values of the various parameters for each scenario.

| Normalized difference | Base Case | Stain. Steel | Aluminum | Thin Probe | Thick Probe | High Soil $a$ | Low Soil $a$ | 1mm Airgap | 5 mm Airgap |
|---|---|---|---|---|---|---|---|---|---|
| 0.1 | 140 | 118 | 92 | 40 | 346 | 118 | 188 | 372 | 1514 |
| 0.05 | 216 | 230 | 168 | 62 | 542 | 166 | 344 | 508 | 1980 |
| 0.01 | 824 | 1040 | 800 | 234 | 2092 | 488 | 1748 | 1070 | 3096 |



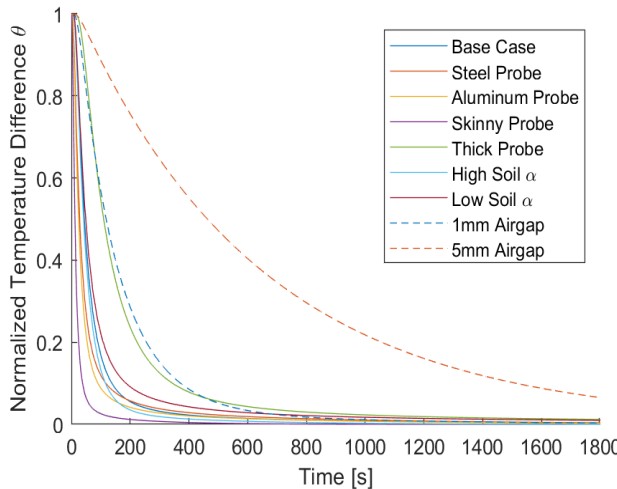

**Figure 4: Equilibrium time for various DTP system geometries and environmental conditions.**

### 3.3 Simultaneous monitoring of snow thickness and snowmelt-infiltration characteristics in a mountainous watershed

Quantifying snow and water distribution in snow-dominated mountainous watersheds is critical for managing downstream water resources and societal services (Viviroli et al., 2007), especially at a time when their functioning is increasingly altered by climate change (Barnett et al., 2005). Climate change and interannual variability in precipitation intensity and surface temperature strongly impact snowpack dynamics and snowmelt timing, stream flow, groundwater recharge, and surface energy balance. A particular challenge to predictive understanding of watershed dynamics and response to perturbations is

monitoring snowpack properties, the timing and magnitude of snowmelt events, and the repartitioning of water into surface and subsurface flow. Such monitoring must be conducted at multiple scales across complex terrains as needed to accurately capture the impact of a large range of gradients in topography, air mass exposure, and vegetation cover on these dynamics (e.g., Lundquist et al., 2019; Strachan et al., 2016). The DTP system has the potential to significantly improve the sampling of these properties and their variability along these gradients.

Two collocated DTP systems, one above and one below the ground surface at a mountainous headwater site in the East River watershed of the Upper Colorado River Basin (Hubbard et al., 2018; Tran et al., 2019), are used to illustrate the DTP data information content on the timing and amplitude of thermal and hydrological processes in the snow and soil columns (Figure 5). The snow DTP system provided temperature with 0.1 m resolution between 0.05 m and 0.85 m above the ground surface and 0.05 m resolution between 0.85 and 1.15 m. The DTP system installed in the soil, next to the snow DTP system,

provided measurements with 0.05 m resolution between 0 and 0.3 m depth, and 0.1 m resolution between 0.3 and 0.7 m depth. Sensor accuracy was 0.06ºC, as probes were deployed before the development of described calibration method. The snow-thickness algorithm is applied to the DTP system placed above the ground surface, and the estimated snow thickness is





compared to the snow thickness measured using a sonic sensor at the Butte Snotel site (https://wcc.sc.egov.usda.gov/nwcc/site?sitenum=380) located 1.5 km away and 300 m higher in elevation. Soil moisture measurements at 0.1 and 0.5 m depth about 1 m away from the DTP probes are used to further evaluate the information contained in the DTP data.

The snow thickness estimated using the DTP system is consistent with the snow thickness observed at the Butte Snotel site (Figure 5). The DTP system captures the main changes in snow thickness linked to snow precipitation, snow melt, and/or snow compaction visible in the Snotel dataset. The earlier timing in snowmelt at the DTP site is explained by the fact that the Snotel station is at higher elevation than the DTP probe. Overall, the main differences between these methods is the lower temporal and spatial resolution of the DTP-inferred snow thickness, which is caused by the use of a 24-hour moving window to estimate snow thickness, the occurrence of days involving very little diurnal fluctuations and/or snow precipitation events, and the 5 to 10 cm spacing between temperature sensors along the probe.

The DTP system placed below the ground surface shows that the soil freezing, which is estimated by extracting the 0°C isotherm from the temperature data, starts in mid-October and reaches 0.4 m depth by mid-November. The first significant snowfall at the end of November increases the insulation of the ground, which leads to a slow decrease in the frozen layer thickness from the bottom (Figure 5a). The soil thawing accelerates in March after the snowpack became thicker and air temperature got warmer. The thawing of soil occurs relatively quickly, likely because of the presence of a relatively dry soil, as indicated by the absence of a clear zero-curtain effect expected in the presence of a large amount of ice and subsequently latent heat absorbed during phase transition. After mid-March, the entirely thawed soil, still covered with snow, remains at an almost constant temperature for about three weeks, with less than 0.01°C change per day (Figure 5c).

The major snowmelt event occurring at the end of March is captured by the aboveground DTP system via the strong decrease in snow thickness as well as by the temperature of 0°C throughout the snowpack. Indeed, once the entire snowpack reaches 0°C, the additional thermal energy entering the snowpack initiates the phase change and water infiltration throughout the snowpack (Figure 5a) (Dingman, 2014; Reusser and Zehe, 2011). The snowmelt water reaching the ground is close to 0°C, while the ground at this location shows a relatively constant temperature of 0.41, 0.65, and 1.38°C at 10, 20, and 50 cm depth, respectively. The snowmelt infiltrating into the ground creates a slight decrease in soil temperature that is apparent in the soil temperature data (Figure 5a) and more clearly identified by looking at the change in the 24-hour average temperature difference (Figure 5c). The change in temperature with depth, which has a different shape than at earlier times during the winter—when heat conduction was dominating heat transfer—is related to the water infiltration. This change in soil temperature is consistent in timing with a soil moisture increase at 10 cm depth around mid-April and ten days later at 50 cm depth. Overall, the high vertical resolution and accuracy of the DTP system and its deployment above- and belowground enabled the observation of snowpack dynamics and its impact on the soil heat (and to some extent hydrological) fluxes at resolution that is not achievable with traditional sensors. While only simple metrics have been extracted here, the snow and soil temperature data gathered with the DTP system establish the foundation for improving the modeling of snow and soil hydrological processes in the future.





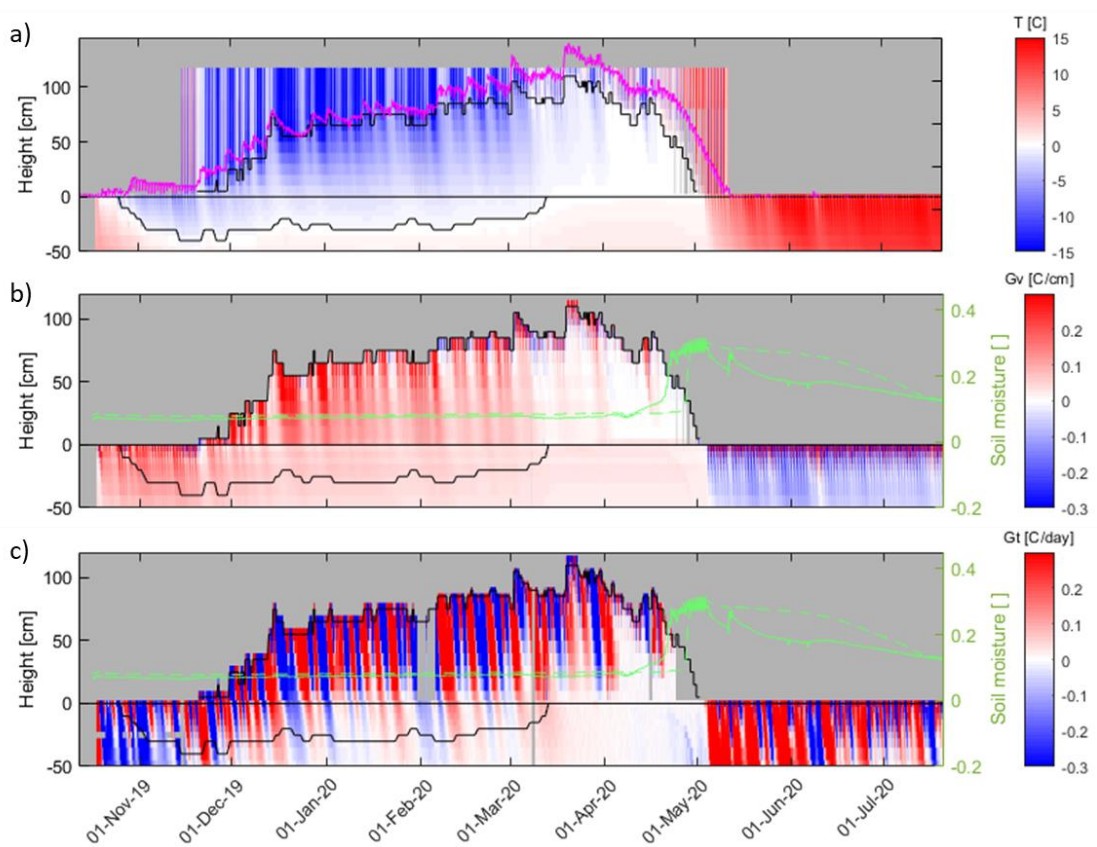

**Figure 5: DTP systems deployed for snow and soil temperature measurements at a site in the East River (Colorado) watershed.**
**The DTP-inferred snow thickness and soil frozen-layer thickness are overlaid on the DTP (a) temperature, (b) vertical gradient at**
**each sampling time, and (c) temporal trend after averaging the time series with a moving 24-hour time window. The pink line**
**shows the snow thickness from Butte Snotel station located 1.5 km away and at a 350 m higher elevation. The solid and dashed**
**green lines indicate the soil moisture at 10 and 50 cm depth, respectively. Color scales have been cropped to the displayed**
**minimum and maximum values in order to improve visualization.**

## 3.4 Monitoring soil temperature, frozen/thawed layer thickness and probe displacement in an Arctic permafrost system

Large uncertainty remains in how northern high-latitude environments will evolve under climate warming, and in particular in how thaw and release of permafrost carbon will be offset by increased vegetation carbon uptake (Jorgenson et al., 2010; Parazoo et al., 2018). Arctic annual average air temperatures between 1971–2017 increased by 2.7°C, at 2.4 times the rate of the Northern Hemisphere average (Box et al., 2019). This change in temperature is complemented with changes in other atmospheric properties, including humidity, cloud formation, rainfall, and snowfall precipitation. One particular challenge involves improving predictive understanding of how permafrost regions transition to unfrozen ground, and disentangling the various controls and their individual impact on the carbon cycle (Jorgenson et al., 2010). Overcoming this challenge requires





improving our capability to measure the soil freeze/thaw depth, the impact of spatially variable temporal shifts in insolation
and insulation on the subsurface temperature, and the water/heat fluxes.

Here, a DTP temperature profile from a site located in a discontinuous permafrost environment along Teller Road (mile 27)
near Nome, Alaska (Léger et al., 2019; Uhlemann et al., 2021) is used to further illustrate the value of the DTP system in
monitoring temperature, frozen layer thickness, and thaw layer depth, as well as to discuss the potential issue of DTP system
displacement relative to the ground surface (Figure 6). The displayed DTP dataset comes from a probe deployed at a location
where the permafrost table is deeper than the bottom of the probe located at 1.05 m depth. The deployed probe provided
temperature with 0.05 m resolution from 0.05 m above the ground surface to 0.25 m depth and with 0.1 m resolution from
0.25 m to 1.05 m depth. The bottom of the frozen layer in the fall and winter time, and the bottom of the thawed layer in the
spring to fall season, were estimated by selecting the deepest sensor with soil temperature below and above the 0°C isotherm,
respectively. The dataset discussion involves an evaluation of the snow depth and air temperature obtained at a nearby
monitoring site (< 1 km) using a sonic-based snow sensor and air temperature sensor, respectively (https://ngee-
arctic.ornl.gov/data/pages/NGA243.html).

Soil freezing, which starts at the end of October before being slowed by a warm event coupled with snow precipitation in
late November, reaches a depth deeper than the length of the DTP probe in early February (Figure 6b). The small amount of
snow (< 30 cm) on the ground favors soil freezing until snow event intensity and air temperature increase in March and
April. Consequently, the ground temperature increases and the temperature of the entire soil column reaches temperatures
slightly below 0°C at the end of April. The soil thawing process starts after the first bare ground day, as indicated by the
diurnal daily temperature variation becoming visible at the ground surface. The soil thawing occurs slowly, with a zero-
curtain effect indicating the presence of wet conditions. The thaw layer thickness increases from mid-May to mid-August, at
which time the thawing occurs deeper than the DTP probe.

The detection of a persistent and negligible time delay in daily minimum temperature between the aboveground sensor and
the underlying nearest sensor indicates the presence of a second sensor above the ground surface, and thus an upward
displacement of about 2.5 cm (+/- 1.5 cm) of the DTP system relative to the ground surface around June 3. Then, there is an
additional displacement around August 20, leading to a 7.5 cm (+/- 1.5 cm) total displacement during the thawing season
(Figure 6c). The developed detection method provides reliable detection of probe movement relative to the ground, though it
does not enable centimeter-scale resolution. Still, the approach allows us to flag the data for lower accuracy and possibly
apply subsequent corrections to the reference depth of temperature data and inferred metrics.

Overall, the high vertical resolution and accuracy of the DTP system enables monitoring of temperature—and related
frozen/thawed layer thickness—in the Arctic environment at a resolution that allows us to disentangle the impact of various
processes on soil warming and changes in hydro-biogeochemical processes. Even with the difficulties in monitoring extreme
environments, the DTP system offers a way to account for various sources of measurement uncertainties and potentially
develop the dense datasets needed to improve predictive understanding of Artic feedback to climate change.

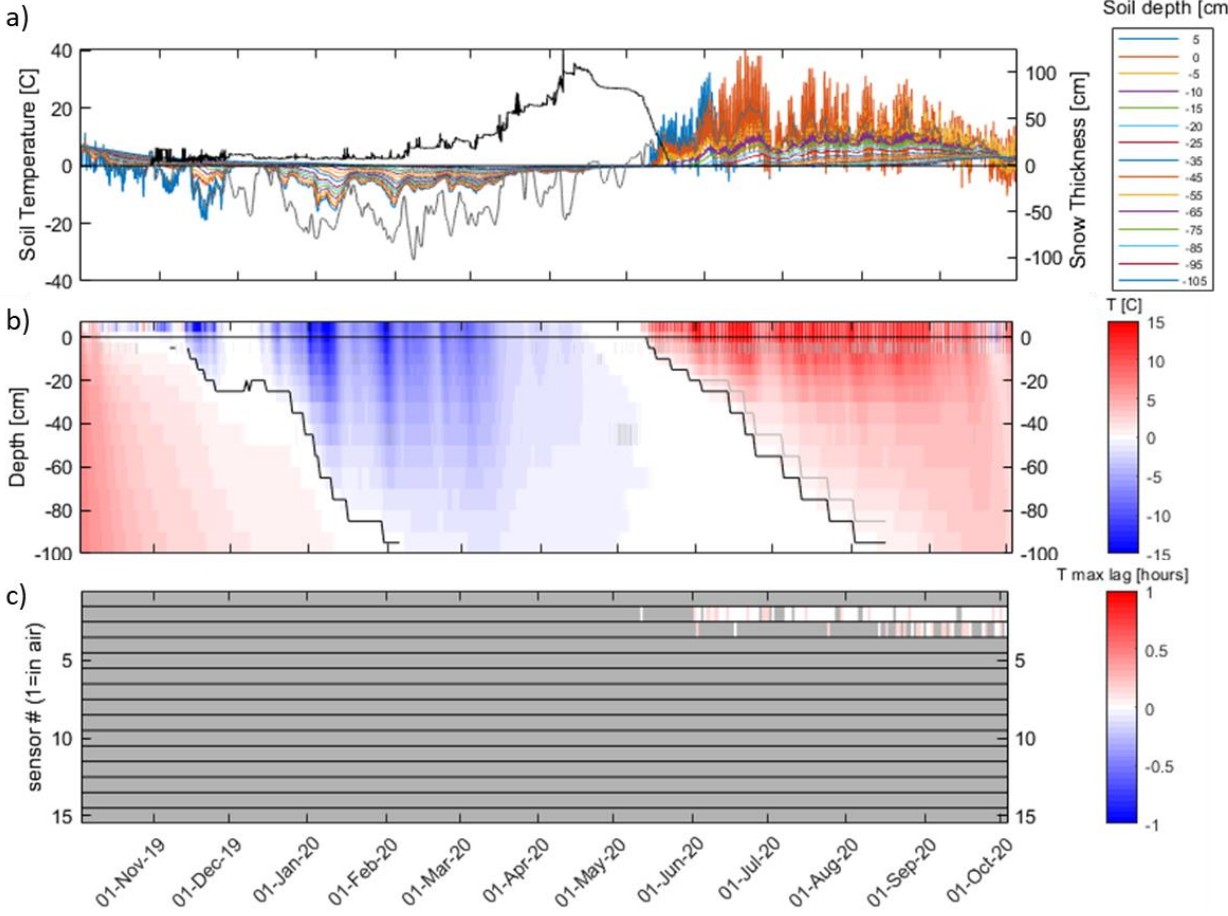

**Figure 6: (a) soil temperature measurements from a DTP system in a discontinuous permafrost environment (Teller Road, mile 27, Nome, Alaska) with sensor from 5 cm above the surface to 105 cm depth, overlaid with the snow depth (black line) and daily average air temperature (grey line) measured at a weather station located about 1 km away from the DTP system; (b) same DTP dataset with black lines indicating the inferred bottom and top of the frozen layer; (c) time delay smaller than 0.25 hour in daily minimum temperature between aboveground top sensor and each sensor along the probe, indicating the presence of an additional sensor positioned above the ground surface due to the an upward displacement of the DTP system relative to ground surface. A ~2.5 cm displacement is flagged around June 3, and then August 20, which together led to a ~ 5 cm shift of the probe relative to the ground surface. The grey line in (b) indicates the thaw layer thickness estimate after correction applied for the June 3 displacement.**

## 4 Discussion

The developed DTP system fulfills numerous requirements for measuring soil or snow temperature with unprecedented lateral and vertical spatial resolution across the landscape. The development and assessment of the DTP system has shown that the use of digital sensors mounted on PCB sections is appropriate for (1) managing a large number of sensors, (2) enabling repeatable measurements and the assessment of measurement accuracy, (3) reaching low production and assembly costs needed for building hundreds of probes, (4) providing flexibility in building probes with various sensor





spacings, length, and packaging, depending on the intended applications. In addition, the development of a custom logger to

communicate with the PCB-mounted temperature sensors offers (1) a compact and low-power solution crucial for limiting the installation complexity and footprint, (2) a low-cost solution compared to other logging options, which is needed for the deployment of a large number of probes, (3) efficient data transfer through Bluetooth LE and other wireless connectivity solutions in the future, and (4) publicly documented hardware and software design that offers control over the entire data acquisition-to-management pipeline.

Our developed calibration approach also enables a reliable assessment of sensor accuracy and provides an additional calibration of the temperature sensors. Results indicate that the digital temperature sensors satisfy the factory-assured, NIST traceable accuracy of +/- 0.1ºC. Moreover, all sensors tested in this study showed an accuracy better than +/- 0.06ºC. The novel calibration approach has also been successful in increasing the sensor accuracy to +/- 0.015ºC. This in-house calibration, along with the factory-assured accuracy, are (to the author's knowledge) unprecedented for digital sensors

deployed in environmental systems, and are relatively close to the accuracy that can be reached with high-accuracy analog sensors and loggers.

Besides the sensor accuracy, numerical simulations of heat transfer in soil and along probes have enabled an evaluation of how probe characteristics and various environmental factors can further affect measurement accuracy. The assessment of measurements errors, though rarely done, informs both the potential and limitations of various methods in capturing small

changes in temperature gradients. Capturing small changes in temperature is critical for estimating fluxes or thermal parameters using physically based models (e.g., Brunetti et al., in prep) or in evaluating processes linked to water-phase changes. Results of the numerical study indicate that, in favorable environmental conditions and a soil diffusivity around 0.5 mm$^2$/s, the use of a ¼'' ID 3/8'' OD diameter plastic probe provides measurements with less than 0.0032% and 250 s in amplitude error and time delay, respectively. Still, results have shown that the potential presence of different surface heating

between the probe and soil surface can significantly affect the measurements in the top few centimeters (Figure 3). In addition, results have indicated that the use of stainless-steel instead of plastic tubing, though implying a decrease in accuracy, can provide similar performance if hammering the probe into the soil precludes the presence of an air gap between the soil and probe.

Importantly, the above probe characteristics have shown impacts on temperature measurements smaller than those owing to

uncertainty in sensor vertical positioning. An error of ±1 cm in positioning a sensor in the soil can lead to an 8.2% and 1460 s amplitude error and time delay, respectively, for a typical soil with thermal diffusivity 0.5 mm$^2$s$^{-1}$. Note that such positioning inaccuracy can occur as a result of either an error in installing a sensor at a precise distance from the ground surface and/or relative to another sensor. The first issue is relevant to all measurement techniques and linked to the difficulty in assessing what is the ground surface (particularly in heavily vegetated landscapes), as well as the potential upward

movement of the sensors relative to the soil surface. The second issue—which is absent in the DTP system, where millimeter precision in sensor spacing is achieved—is conspicuously present in other types of measurement techniques, including fiber-optic based methods or individual point-scale sensors deployed at different depths (e.g., Steele-Dunne et al., 2010). Overall,



while the DTP equilibrium time and measurement accuracy is not as high as theoretically achievable with sensors in direct and tight contact with the soil, the numerical experiments enable a clear assessment of the advantages and limitations of
various measurement strategies and devices.

The deployment of the DTP system for monitoring snowpack thickness has confirmed the results from earlier studies (e.g., Reusser and Zehe, 2011). In particular, this study confirms that a vertically resolved temperature probe can be used for daily estimation of snow thickness with an accuracy close to the spacing of the temperature sensors. In addition, capturing temperature throughout the snowpack is crucial for detecting the onset of snowmelt events and water infiltration into the soil
or potential surface water runoff. Although it is beyond the scope of this particular study, the acquired snow temperature data can be potentially used for estimating soil thermal parameters (e.g., Oldroyd et al., 2013), soil water equivalent (SWE) (Sturm et al., 1997), snow insulation effects on ground temperature (Jafarov et al., 2014), snow accumulation, densification and distribution, water infiltration, surface flow, or spring floods. An additional advantage of the DTP snow probes is their low spatial footprint and suitability for deployment in steep hillslope and at-risk locations. This advantage comes with the
caveat that the DTP-inferred snow thickness has lower resolution and accuracy than temperature-corrected sonic-based sensors, and that its overall value is limited where snowpack is generally thicker than a few meters, as the costs of the probe scale up with the number of temperature sensors. Though the DTP system is not intended to replace sonic-based sensor and intensive sites, it opens the door for dense networks of snow depth, temperature, and potentially SWE estimates at watershed scale, where predictability is still limited owing to the complexity and cost of capturing precipitation heterogeneity and
widely variable hillslope-scale heterogeneity, as well as a wide range of energy dynamics. Capturing both the local and larger-scale snow characteristics is critical in developing statistics on the different coupling of landscape and environmental factors, and enable advances in the understanding of watershed aggregated snow and water dynamics.

Besides snow temperature, there is a broad range of applications of the DTP system for monitoring soil temperature, inferring metrics (such as thaw layer thickness, frost layer thickness, zero-curtain and thermal parameters), informing on heat
and water dynamics, and validating thermo-hydrological models. As a simple example, this study reported on the use of the DTP system to monitor the frozen layer thickness and the thawing process in a mountainous and Arctic environment. Results show that the 5 to 10 cm spacing between temperature sensors along the DTP system is adequate to reliably track the freezing and thawing front, with a spatial resolution that has been rarely obtained (Cable et al., 2016; Léger et al., 2019). At the same time, the system provides a unique capability to duplicate temperature measurements at a large number of
locations. Additionally, the characteristics of the developed DTP system promise to improve the quantification of water flow (Hatch et al., 2006; Irvine et al., 2020; Tabbagh et al., 2017), ground heat flux (Hurwitz et al., 2012), and soil thermal parameters (Nicolsky et al., 2009; Tran et al., 2017).

Similarly, the system has the potential for popularizing single-time or sporadic mapping of soil temperature across the landscape for various purposes, including the delineation of near-surface permafrost (Léger et al., 2019), the identification of
temperature hot spots or geothermal areas (Lubenow et al., 2016), the detection of temperature anomalies linked to the presence of shallow aquifers or high water content (Cartwright, 1968), or the delineation of thermal regimes—indicators of





various soil hydro-biogeochemical fluxes. Indeed, the DTP system developed in this study is (to the author knowledge) the first system that provides the ability to efficiently install DTP systems for a short period of time (e.g., 30 min) and move them across the landscape at a pace that can enable surveys of soil temperature at hundreds to thousands of locations within a

short time period. Such surveys have remained limited, presumably because of the lack of equipment with an adequate trade-off between the acquisition depth needed to minimize land surface boundary impacts to the extent needed to identify a thermal anomaly, as well as adequate sensor accuracy, vertical resolution and total cost (incl., material, acquisition, data management). The development presented in this study responds to challenges expressed in several studies that have either relied on conventional thermocouple probes (≤25 cm) (e.g., Leon et al., 2014; Lubenow et al., 2016; Price et al., 2017), or

developed their own acquisition devices with limited duplicability due to cost (Hurwitz et al., 2012; Léger et al., 2019). Though the advances presented in this study are important for improving monitoring or mapping of soil temperature across the landscape, several challenges remain. For example, upward movement of the DTP system relative to the soil surface, which can occur because of soil thermal and mechanical processes or interaction with animals, can influence soil-temperature-measurement accuracy over time. While the DTP fine vertical resolution and the upward-movement-detection

approach developed in this study enable satisfactory detection of upward displacement, more complex deployment strategies and algorithms have the potential to improve such estimates. Note that frost jacking or the impact of soil frost and thaw settlement on temperature sensors is not only related to the DTP system, but also to buried individual sensors (Johnson and Hansen, 1974). Overall, the developed DTP system is extensible to a wide range of applications and sufficiently modular to facilitate future developments.

**Conclusion**

This study aimed at developing a low-power and small-footprint DTP system providing vertically dense and high-accuracy temperature measurements at a total cost that would enable its deployment in a substantial number of locations, as needed, to improve the multiscale observation and understanding of environmental system functioning—in particular snowpack and soil thermal and hydrological dynamics. The developed DTP system and our assessment of it have demonstrated its potential

for measuring soil temperature with unprecedented vertical resolution, high accuracy, and low cost, while minimizing physical footprint and energy consumption. Also, this study shows that the developed system provides flexibility in using various types of housing (depending on project goals and environmental requirements), and offers simplicity in downloading and managing data. To our knowledge, it is the first time that temperature data are gathered with such high spatial resolution to capture changes in snow thickness and freezing/thawing layer thickness. We anticipate that the datasets acquire with this

system will be crucial in improving the estimation of thermal parameters and possibly flow across watershed scales, which both benefit from high-resolution and high-accuracy data. These advances are particularly critical for improving our understanding of the various timing and intensity of snowpack and soil thermohydrological dynamics in heterogeneous environments. We expect that the improved monitoring data and scientific insights developed from the data will greatly

improve the predictive understanding of the heat and water fluxes in snow and soil, which is essential for improving water resources and carbon cycle assessment and management.

The DTP system development and accuracy assessment presented in this study is an important step toward deploying large numbers of sensors, as part of a system optimized with regard to environmental monitoring objectives, emphasizing accuracy, resolution, repeatability and low equipment and measurement costs. The development of hardware and software, and their release into the public domain, is similarly important to ensure knowledge transfer and future developments. Here as a first step toward this objective, we presented the capabilities of a DTP system that uses TMP117 sensors and a custom logger design described in detail. The level of detail that has been provided on the system design assures the repeatability of experiments and the development and advancement of future DTP systems, using the same or improved components. The DTP system opens new possibilities for observing thermohydrological processes at numerous locations, and provides the flexibility for adapting it to applications not discussed in this study, including in-stream deployment. Ongoing additional developments include a Python-based numerical framework and toolbox for automated extraction of metrics and estimation of temperature-related processes; the addition of LoRa connectivity for real-time transmission of data from hundreds of nodes over several kilometers to data hubs; and the incorporation of additional low-cost and low-power sensors to the system.

### Data availability

The data presented in this study are available from the NGEE Arctic data portal at https://doi.org/10.5440/1819363.

### Author contribution

BD, SW, JL, IS, FA designed the DTP acquisition strategy. BD, FA, SP, JF, SW carried out the development of the DTP hardware and software. JL performed numerical simulations. BD and PM developed the calibration approach. BD, CB and CW worked on algorithms development. BD, PM, JL, SF, SW, SU, IS, CU and JP participated in building the DTP systems and/or data collection. RB provided meteorological datasets. BD prepared the manuscript with SW, JL, PM, CW, SU and SH. All authors contributed to the study and approved the final version of the paper.

### Competing interests

The authors declare that they have no conflict of interest.



**Acknowledgments**

This material is based upon work supported primarily by the Next Generation Ecosystem Experiment (NGEE-Arctic) and secondly by the Watershed Function Scientific Focus Area, both funded by the U.S. Department of Energy, Office of Science, Office of Biological and Environmental Research under Award Number DE-AC02-05CH11231. We acknowledge the assistance of Berkeley Lab's Geoscience Measurement Facility (GMF) for providing help in early prototyping of the DTP system.

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
