# Peer review of "A distributed temperature profiling system for vertically and laterally dense acquisition of soil and snow temperature"

_The Cryosphere, 2021_

## Author Response (AR1)

Dear Editor,

Thank you for organizing the review process and for your guidance and suggestions. We addressed all the reviewers' comments. In particular, we improved the organization of the introduction and discussion section, provided more details where suggested by the reviewers, and enhanced the content and quality of the figures and captions.

**RC1**: 'Comment on tc-2021-292', Anonymous Referee #1, 22 Oct 2021

*The manuscript "A Distributed Temperature Profiling System for Vertically and Laterally Dense Acquisition of Soil and Snow Temperature" by Dafflon et al. is a technical description of high-density temperature measurement (Distributed Temperature Profiling; DTP) system of ground surface and subsurface. The newly developed DTP system has an unprecedented measurement capacity with low cost, high resolutions, easy data retrieval, and flexibility in the sensor configuration. The authors also perform numerical experiments to assess potential errors in practical scenarios of sensor installations. Finally, they demonstrated the performance of the new DTP system in two field case studies. I agree that there will be numerous applications using this densely obtained temperature, which will certainly lead to new understandings in various areas of environmental studies. This DTP system will be breakthrough instrumentation in environmental monitoring when it becomes readily available for the public. The overall quality of the study is excellent, but there is a space to improve the presentation of the contents. I have minor comments to improve readability and attention from the research communities.*

Dear Reviewer,

Thank you for your comments regarding the quality of the manuscript and study. We very much appreciate your review suggestions, which greatly helped us to improve this manuscript.

*The manuscript structure is not consistent with the chaptering. This paper would not fall into a traditional structure because it contains a section of numerical simulation and two field case studies. Although Sections 3.2, 3.3, and 3.4 are in the Results chapter, they have their introduction and methods before the result descriptions. If authors want to follow the conventional chaptering, they should be consistent. Or, they could re-organize the chapters and sections separately for each different component.*

We agree this manuscript does not have a traditional structure, primarily because it includes a numerical study, and two field cases. We think that introducing the site and data for the field cases before the numerical study would not make the reading and understanding of the paper easier. Though we recognize there is no perfect solution for organizing the manuscript, we decided to follow recommendation of Reviewer RC2, who suggest that we do not change the structure of the paper. We are inclined to modify the structure of the paper if the editor and reviewers all agree.

*The Introduction explains various aspects of the importance and applications of DTP. However, I think this Introduction could be more concise in explaining former application in the dense measurement of temperature, and technical challenges in the previous studies that could be overcome in this study should be more focused on. Although the authors summarized shortcomings of an earlier prototype of the DTP system (Leger et al., 2019), the readers would like to know a summary of individual limitations and shortcomings of the previous attempts of DTP referred to in Introduction.*

We improved the introduction by making it more concise. In particular, we reduced the discussion of former applications. Also, we provided more details on the limitations and shortcomings of Leger et al (2019).

*The sentence of the research objective starting from L113 is tediously long, and I could not understand it smoothly. And this doesn't seem to match the content of this paper. "to improve our predictive understanding of ..." is also the objective of this study? I think this is suspended from the previous phrase "to design and develop," but this sentence is confusing.*

We agree this sentence is too long and confusing. We removed the second part of the sentence and improved the paragraph.

*L146: "digital temperature sensors" It would be helpful to explain in more concrete wordings (e.g., semiconductor-based IC?) for readers who are only familiar with thermistors, RTDs, and thermocouples. Every sensor output could be digital. Provide a long-term trending of the sensor quality.*

Thanks for the suggestion. We improved the description of the sensor accordingly to the reviewer's comment. It can be noted that there is currently no assessment of the long-term trending at room temperature and under various operating conditions. Based on the manufacturer documentation, long-term stability was determined using accelerated operational life testing at a junction temperature of 150°C (with an unknown number of sensor). The range of observed differences between pre and post 1000 hours at 150°C was reported to be +-0.03 ºC. This value corresponds to an unspecified lifetime at room temperature.

*L172-3: This is beneficial information. Could you also provide the battery life with Bluetooth connectivity (always on / occasional on) if possible?*

We clarified that Bluetooth is always on and provided the calculation of the battery life.

*L260: soil and temperature à soil temperature?*

Corrected. Thanks.

*L264: frost-affected à frost-susceptible*

We followed the reviewer's suggestion.

*L337: I was confused with "measured" and "true" temperatures. What were compared? The analytical solution with simulated temperature by your numerical model? Or "true" measured temperature in the field??? Hypothetically measured temperature?*

We are comparing soil temperatures simulated with and without the numerical representation of the probe characteristics We have now clarified this in the manuscript.

*Figure 3: Please use the same term in (b) graph and caption. Phase shift vs. time delay in amplitude.; Please consistent use of a or alpha for diffusivity in Table 1 and Figure 3.*

We agree there were some inconsistencies between the figure and the captions. We addressed all of them.

*Section 3.3: This section describes the overall performance of the temperature and snow depth measurements, but snow depth estimation could be more focused. This study used a different algorithm using temperature gradient from previous studies with standard deviation differences. Why did you choose the new algorithm, and what was the performance compared to the previous algorithm? Figure 5b could be more explained to describe the result of snow thickness calculation and its reliability.*

We improved the discussion of the snow depth estimation, and further discuss the difference with algorithms used in previous studies. In our study the use of the temperature gradient enabled better results than using the standard deviation difference. However, we recognize that the performance of various algorithms may be case dependent due to the variable dynamics (in diurnal cycle, temperature during snow melt, etc) occurring over time and across various regions (e.g., in Arctic vs in Colorado). Thus, we do not intend to claim our algorithm work better than others. We added a sentence clarifying that further evaluation and development of algorithms for snow depth estimation may benefits from testing them on a large range of conditions and sites.

*Figure 5: Unit for Soil moisture missing; The abbreviations for the title of color bars on the right of the figure should be explained in the capture. The color pallet, especially for Gv (b), should not be cropped because the readers like to see how prominent the boundary of the Gv between in air and in the snow was.*

We agree there were some inconsistencies between the figure and the captions. We addressed them. We decided to keep the color pallet cropped (and mentioning it in the caption) as it increases visibility of small-scale variations.

*Section 3.4 also has the same structure and nature in describing the other case study.*

We ensured Section 3.3 and 3.4 have the same structure.

*What material was used for the DTP tubes in the two case studies, plastic or metal?*

We used plastic for both field studies. We clarified this in the text.

*Figure 6: Is the snow depth derived from the DTP system or from ultrasonic? Are there data gaps in +5cm temperature? If so, please describe the gaps too.*

The snow depth in Figure 6 is from a sonic sensor (we now mention it in the caption). The temperature at +5cm is not visible on the figure after June 3 because it is getting overlapped by temperature data from deeper sensors shifted above the ground-surface (due to the upward movement of the probe). We clarified this in the caption.

*4 Discussion: this chapter is mostly a summary of the previous chapters. Potential usages of dense temperature measurements enabled by the DTP system are repeatedly described here after the descriptions in 1 Introduction. They could be merged, at least for some portions.*

We agree with the reviewer. We removed several sentences to avoid redundancies between the introduction and the discussion.

*Instead, the discussion of the system's advantages and limitations, which is currently performed only in the last several sentences. They could be sufficient, but if possible, please add discussion on pitfalls/difficulties in detecting temperature boundaries to decide freeze-thaw or snow-air interfaces.*

We added a sentence discussing potential difficulties and limitations that may be encountered when estimating snow thickness.

**RC2**: 'Comment on tc-2021-292', Michael Prior-Jones, 25 Oct 2021

*My expertise is principally in electronic engineering, so I will make detailed comments on that aspect of the article, and will make only limited comment on the other aspects. This paper describes the development and field evaluation of a new temperature sensing instrument for measuring temperature profiles within snow or soil. The authors describe the development of the electronics, use thermal modelling to estimate the instrument's performance, and develop a new approach to calibration which allows the digital temperature sensors used to perform at an accuracy in excess of their datasheet values. Finally they present results from field studies which demonstrate the potential value of this new instrument for studies of snow, ice and frozen soil. Broadly this looks like a solid piece of development work and contains a lot of useful insights as well as promising considerable advances in the field once the instruments are deployed in large numbers.*

Dear Reviewer,
Thank you for your comments regarding the quality of the manuscript and study. We very much appreciate your review suggestions, which greatly helped us to improve this manuscript through revision.

*Brief points of style and structure: in my view the introduction is rather longwinded and could be cut down to focus on the important points, which are that temperature profiling is an established technique for studies of both and snow and soil, and that existing methods are cumbersome and expensive.*

We cut down the introduction and improved it.

*I note that one other reviewer has commented on the structure with regard to separating methods and results. My feeling is that an instrument-development paper is easier to read if you have the description of the engineering design rationale up front (as you have done) and then present the subsequent modelling, lab tests and field tests into separate chapters for each piece of work, each with its own methods, results and discussion. However, that's not normally how a scientific paper is presented and so I leave that as a merely a suggestion to the authors and editor!*

We agree with the reviewer. We kept the description of the instrument and related modeling and performance assessment upfront and we introduce the field studies later in the paper.

*Lines 152-3: I'm presuming that the D-type flip-flops are arranged so that the whole length of the probe appears like a shift register, and then the logger clocks a single logic "1" down the chain to*

*enable each IC in turn. I think this would benefit from further explanation or a diagram (perhaps as a Supplemental) as it's not immediately obvious. It may also be worth pointing out that TWI and I²C are the same thing, as some people may have heard of one and not the other.*

We have now provided more details in the text to make this clear.

*Line 166: please give the part number for the RTC used.*

Done.

*Line 180: it would be helpful to give an indication of the amount of memory in bytes used to store each measurement. It's possible to back-calculate this from your description of the Bluetooth data transfer speed, so why not state it explicitly? This will come in useful for anyone wanting to connect the instrument to a satellite modem for real-time reporting from a remote region.*

We modified the manuscript to mention it explicitly.

*Line 187: please give a manufacturer (and ideally a model number) for the CAB tube used.*

We provided more details to ensure the reader can easily find/order such tubes, that are custom made by various companies. We do not want to include manufacturer names in the manuscript to not advantage one or another company. We can provide this information to the reader if contacted.

*A more general point in this section: I'm not sure what the authors' position on the intellectual property in the design is here, but if IP considerations allow, it would be wonderful if the whole design could be published open source (electronics schematics, PCBs layouts, firmware, etc) alongside this paper. If that's not possible, maybe a complete bill of materials listing all the parts and their sources as a supplemental item? Neither of these are showstoppers for the paper but they would provide valuable information to anyone trying to replicate their work.*

We agree with the reviewer that an open source release would be great. We are currently working on having all the information released. We added now a list of all the components (serial number and manufacturer) used to build the system in the data archive linked to the manuscript. The PCB layout and the logger firmware will be archived at a later time. The readers can contact the author to obtain more details.

*Line 237: If I've understood this correctly, the calibration protocol is:*

- *Set the data loggers running*
- *Chill to -5C over 12-24 hours to ensure everything is fully frozen*
- *Transfer to a +3C incubator (or are you simply changing the setpoint on the -5C one?) and allow to warm up*
- *Wait until everything has come to equilibrium and then look at the results*
- *Look for the inflection in the time-temperature graphs, average it and then define that as the offset from zero*

*Again, it would be good to have this clearly specified. It does become slightly clearer once you've seen the results section over the page.*

We improved the description of the calibration protocol.

*Figures 5 and 6: I found these both quite hard to read, especially on a printout. In particular, the thin green line used for soil temperature in figures 5b and 5c are almost invisible, especially for the dashed line.*

We improved the figure.

**RC3**: 'Comment on tc-2021-292', Anonymous Referee #3, 02 Nov 2021

*This research paper presents the development and validation of a low-cost, low-powered Distributed Temperature Profiling (DTP) system to be used for soil and snow temperature monitoring. Details of the DTP system is presented as well as a novel calibration approach on how to increase the factory-assured sensor accuracy from $\pm 0.1$ °C to $\pm 0.015$ °C. They also presented results from two field sites where the first site demonstrated the capability of using the DTP system for understanding snowpack and how it affects water resources in a mountainous location, and the other site was focused on understanding soil properties for a carbon study. This paper includes a detailed description on how to build this system and how to improve the accuracy of the sensors used in this study. I think that the general science community would benefit from this paper. Here are some specific comments that could improve this paper.*

Dear Reviewer,
Thank you for your comments regarding the quality of the manuscript and study. We very much appreciate your review suggestions, which greatly helped us to improve this manuscript through revision.

*Line 96: The period is missing.*

Corrected. Thanks

*Line 200-215: This paragraph would benefit from adding how long time it takes to build a DTP system. How long does it take to build a 1.2 m long probe? How long does it take to program the logger? Is the program shared somewhere? If someone wants to duplicate this system, it would be beneficial if the program can be found at a shared site or perhaps in a supplemental?*

We added a sentence to describe the time needed for assembling the probes. We added a list of all the components needed to build the system in the data archive linked to this manuscript. The logger layout and firmware will be released at a later time. We suggest people interested in building this system to contact the authors.

*Lines 234, 359, 375: inches are used throughout the paper for the OD. Consider having that measurement in parenthesis and use mm instead seeing as the paper is in SI units. There might be other places throughout the paper where inches are listed. If you decide to change, stay consistent.*

We agree and modified accordingly.

*Line 239: There is a space after ° in "°C"*

Corrected.

*Line 328 and 331: replace the degree symbol to be consistent with °C everywhere else in document.*

Done.

*Line 386: Here the units are given as mm2/s and Wm-1K-1. This not how the units are written throughout the text. Change to stay consistent. Should it be soil diffusivity "α" as in the alpha sign (I couldn't add the alpha sign)? α (alpha sign) is used in the legend of Figures 3 and 4. Should this be in italic? Same for thermal conductivity "k"?*

We improved consistency.

*Line 423: Snotel should be "SNOTEL". Please change everywhere in the document.*

Done.

*Line 424: The distance and elevation difference from location of DTP and SNOTEL site could be drastically different. Authors should mention that the snow depth measured at a 1.5 km distance away will not be exactly the same due to elevation difference and location. It does show similar patterns between the two but there should be a note somewhere about this.*

We modified the manuscript to mention it.

*Line 462: Is it 350 m higher elevation or 300 m as written on Line 424?*

It is 350 m higher. We have now improved consistency.

*Line 539: Remove an s from "measurements" in "measurements errors"*

Done

*Line 551: write "/s" rather than "s$^{-1}$"*

Done.

*Line 566: Is it a true statement that temperature can be used to determine SWE? The 1997 Sturm paper doesn't mention SWE I don't think. Is it supposed to be snow thermal conductivity?*

We agree our sentence was not clear as they are several steps involved in potentially estimating SWE, including estimating thermal parameter and then inferring density. We removed this statement.

*Line 568: should it be "and/or spring floods"?*

Corrected.

*Table 1: Add a space before the "3" in (5,3) (column "Probe diameter (mm) (OD, ID)")*

Done.

*Table 2: Should "a" be "α" (as in the alpha sign). Space after "1" in "1mm"*

Corrected.

*Figure 2: What is the time scale in a)? Remove "Datetime" and add minutes/hours or whatever the time period might have been. Just a date doesn't really show how long the calibration was performed.*

We agree. We modified the figure to show the time in minutes.

*Figure 3: In the legend "Skinny" is used. Should that be replaced with "Thin"? You could remove "error" from the legend seeing as that is clear from the x-axis. Wait, it isn't an error for b is it? So just removing "error" from legend should work. Also, it is "Stainless Steel" and not just "Steel" right?*

We are now using the word "Thin". We removed "error" from the legend. We now mention "stainless steel".

*Figure 5: I find this figure a bit unclear. The green lines are really hard to see so I suggest changing the color of the lines. Because a, b, and c, are all different units, should the colors be different? What does G stand for? Gradient? This need to be added to the figure caption. The black line is inferred snow depth and frost depth? Mention that in caption. Add the degree sign to the legends. Should it be "snow depth" rather than "snow thickness? This comment is for the whole manuscript.*

We improved the figure accordingly to the reviewer's suggestions. We now mention snow depth through the entire manuscript.

*Figure 6. Maybe make snow depth thickness and air temperatures using a thicker line so that it can be easier distinguished? I suggest removing negative snow depth in a. Some of the "-100" is cutoff in b. Add degree symbol before "C".*

We improved the figure accordingly to the reviewer's suggestions. Thanks for your suggestions and careful reading of the manuscript.

**CC1**: 'Comment on tc-2021-292', Achut Parajuli, 28 Oct 2021

*In this article entitled "A Distributed Temperature Profiling System for Vertically and Laterally Dense Acquisition of Soil and Snow Temperature", the authors have presented an inexpensive DTS system to monitor snowpack temperature profile, soil temperature profile as well as snow depth.*

*However, similar research is presented by Lundquist & Lott (2008) (DOI: :10.1029/2008WR007035) using the exact same DTS device. A similar approach has been adopted in Sodankylä, Finland (see DOI: 10.1029/2020MS002144). More recently, a similar snow temperature profiler has been used for various application: derivation of snowpack cold content (DOI: 10.5194/tc-2021-98) and in temperature index snow model (DOI: 10.3390/w12082284). Also, a similar approach has been adopted to understand the permafrost dynamics (DOI: tc-13-2853-2019). Therefore, the applicability of snow temperature profiler (DTS system) is immense. However, I believe this is not a novel work.*

Dear Achut,

I carefully read the studies you are referring to and I think our manuscript is far from being "similar" to any of these studies. Below, I clarify the differences between our study and the studies you mentioned. I hope it will help you to re-evaluate differences and similarities between our and other studies. I stay available for answering any other question.

Lundquist & Lott (2008) (DOI: :10.1029/2008WR007035): The comment wrongly claims we use "the exact same DTS device". Lundquist & Lott (2008) study, which we referred to in our introduction, used self-recording temperature sensors (Maxim iButtons) placed on or a few cm below the ground surface. Our method is very different as we measure temperature at multiple heights along a vertical profile with a single logger, and extract different information than possibly done with the device they used.

Day et al. (2021) study with data from *Sodankylä, Finland* (DOI: 10.1029/2020MS002144): Day et al. (2021) paper contains a vertically-resolved profile of snow temperature. Snow temperature profiles have been measured in several studies (e.g., Oldroyd et al., 2013; Reusser and Zehe, 2011) as described in the introduction of our manuscript. Overall, the topic of Day et al. (2021) study is very different from our study which aims at developing a temperature probe and logger to improve our ability to perform vertically-resolved temperature measurements at numerous locations.

Parajuli et al. (2020) (DOI: 10.3390/w12082284) and Parajuli et al. (2021) (DOI: 10.5194/tc-2021-98). Parajuli et al. (2021) are using a snow temperature profiler described in their paper by the following sentence: "Inspired by Lundquist and Lott (2010), we deployed an automated snow-profiling station at each location, composed of 18 T-type thermocouples vertically spaced 10 cm apart". This approach constitutes one more application of snow temperature profiling (e.g., Oldroyd et al., 2013; Reusser and Zehe, 2011). We do not think that measuring a snow temperature profile (as many other study) is an argument to say that these studies are "similar" and thus not novel. In addition, we can add that Parajuli et al. (2021, under review) seems to recognize the limited amount of snowpack datasets by mentioning in the introduction that "The exact determination of *CC (Cold Content)* requires direct observations of the snowpack temperature, density, and depth, usually collected from manual snow surveys. As manual collection is tedious and demanding, few datasets that describe snowpack *CC* are available". Improving instrumentation for measuring temperature over time and space is what we intended to do in our study.

Leger et al. (2019) (DOI: tc-13-2853-2019): It is an earlier study that we published. We discussed in the introduction of our manuscript how the new system is different from the older one. In short, we went from a Raspberry Pi controlled probe that could not be used for long-term monitoring (due to power consumption, reliability and bulkiness) to a system that is deployable at numerous locations at an acceptable total cost. We clarify this in our manuscript.